# IL-21/IL-21R Promotes the Pro-Inflammatory Effects of Macrophages during *C. muridarum* Respiratory Infection

**DOI:** 10.3390/ijms241612557

**Published:** 2023-08-08

**Authors:** Shuaini Yang, Jiajia Zeng, Wenlian Hao, Ruoyuan Sun, Yuqing Tuo, Lu Tan, Hong Zhang, Ran Liu, Hong Bai

**Affiliations:** Key Laboratory of Immune Microenvironment and Disease (Ministry of Education), Department of Immunology, School of Basic Medical Sciences, Tianjin Medical University, Tianjin 300070, China; nier1998@163.com (S.Y.); zjiajia814@163.com (J.Z.); 16622393299@163.com (W.H.); sunry0609@163.com (R.S.); tuoyuqing2021@163.com (Y.T.); tanlu@tmu.edu.cn (L.T.); zhanghong0621@tmu.edu.cn (H.Z.); ranliu@tmu.edu.cn (R.L.)

**Keywords:** chlamydial infection, IL-21/IL-21R, macrophage, polarization, inflammation

## Abstract

Interleukin-21 and its receptors (IL-21/IL-21R) aggravate chlamydial lung infection, while macrophages (Mφ) are one of the main cells infected by chlamydia and the main source of inflammatory cytokines. Therefore, it is particularly important to study whether IL-21/IL-21R aggravates chlamydia respiratory infection by regulating Mφ. Combined with bioinformatics analysis, we established an IL-21R-deficient (IL-21R^−/−^) mouse model of *Chlamydia muridarum* (*C. muridarum*) respiratory tract infection in vivo, studied *C. muridarum*-stimulated RAW264.7 by the addition of rmIL-21 in vitro, and conducted adoptive transfer experiments to clarify the association between IL-21/IL-21R and Mφ. IL-21R^−/−^ mice showed lower infiltration of pulmonary total Mφ, alveolar macrophages, and interstitial macrophages compared with WT mice following infection. Transcriptomic analysis suggested that M1-related genes are downregulated in IL-21R^−/−^ mice and that IL-21R deficiency affects the Mφ-mediated inflammatory response during *C. muridarum* infection. In vivo experiments verified that in IL-21R^−/−^ mice, pulmonary M1-type CD80^+^, CD86^+^, MHC II^+^, TNFα^+^, and iNOS^+^ Mφ decreased, while there were no differences in M2-type CD206^+^, TGF-β^+^, IL-10^+^ and ARG1^+^ Mφ. In vitro, administration of rmIL-21 to *C. muridarum*-stimulated RAW264.7 cells promoted the levels of iNOS-NO and the expression of IL-12p40 and TNFα, but had no effect on TGFβ or IL-10. Further, adoptive transfer of M1-like bone marrow-derived macrophages derived from IL-21R^−/−^ mice, unlike those from WT mice, effectively protected the recipients against *C. muridarum* infection and induced relieved pulmonary pathology. These findings help in understanding the mechanism by which IL-21/IL-21R exacerbates chlamydia respiratory infection by promoting the proinflammatory effect of Mφ.

## 1. Introduction

Chlamydiales are Gram-negative, obligate intracellular bacteria that undergo a unique biphasic developmental cycle in eukaryotic hosts [1]. The major human and animal pathogens are *Chlamydia trachomatis* (*C. trachomatis*), *Chlamydia pneumoniae* (*C. pneumoniae*), *Chlamydia psittaci* (*C. psittaci*) and *Chlamydia suis* (*C. suis*), among which, *C. trachomatis* is a major cause of ocular infections, specifically trachoma, as well as sexually transmitted diseases [1]. The chlamydia strain used in this study is *Chlamydia muridarum* (*C. muridarum*), which was originally isolated from laboratory mice and hamsters and primarily used as infection inoculum in mouse models for female genital and respiratory tract infections [2,3]. Previous studies have demonstrated that the Th1-type immune response, characterized by IFN-γ production, plays a pivotal role in conferring resistance to chlamydia infection [4,5]. Th17 cells could enhance dendritic cells (DC) function through secretion of IL-17 and promote the protective Th1-type immune response to participate in anti-chlamydial protective immunity [6]. In most cases, an efficient immune response is mounted involving recruitment of innate immune cells and eventual clearance of the infection. As such, professional phagocytes, primarily macrophages and neutrophils, act as gatekeepers to contain and resolve infection [7,8].

Macrophages (Mφ) are professional phagocytes capable of engulfing and processing foreign materials, dead cells, and debris. They play important roles in the innate immune response as well as activate the adaptive immune response by presenting antigens of phagocytosed pathogens as antigen-presenting cells, leading to microbial death [9,10]. Pulmonary macrophages, including alveolar macrophages (AMs) and interstitial macrophages (IMs), are important innate immune cells involved in the normal physiological functions of lung tissue and some acute and chronic lung diseases, such as *Mycobacterium tuberculosis* (*M. tuberculosis*) infection, *Staphylococcus aureus* (*S. aureus*) infection and fibrosis [11,12,13,14,15,16]. Mφ polarization involves the adoption of a classically activated M1 (pro-inflammatory) state in response to IFN-γ and/or LPS, and the acquisition of an alternatively activated M2 (anti-inflammatory) state following exposure to IL-4 and IL-13 [9,17,18]. The M1/M2 balance is crucial for the maintenance of Mφ phagocytosis and immune function [19]. For instance, the pro-inflammatory response of AMs infected with *M. tuberculosis* is enhanced, whereas M2-like AMs are more permissive to bacterial growth than M1-like IMs [18]. *Enterococcus faecalis* infection of bone marrow-derived stem cells during differentiation into Mφ induces an atypical M1-like phenotype associated with intracellular bacterial survival. And this atypical M1-like phenotype is retained even upon stimulation with growth factors that normally trigger their development into M2 [20]. In intracellular *S. aureus* infection, high levels of reactive oxygen species (ROS) induce antibiotic tolerance through an M1-like pro-inflammatory Mφ response, but modulating the Mφ response to an anti-inflammatory M2-like state facilitates resolution of established *S. aureus* skin and soft tissue infections, and bacteremia [21].

As a member of the IL-2 cytokine family, IL-21 is synthesized by activated T cells, including Th17 cells, activated NKT cells, and T follicular helper cells [22,23]. IL-21 signals through its IL-21 receptor (IL-21R), which is composed of CD132 and IL-21Rα subunits. The expression of IL-21R is predominantly confined to immune cells, including T cells, B cells, natural killer cells, Mφ, and DC [9,24,25]. Thus, IL-21 exhibits diverse functions in various diseases. In the context of *M. tuberculosis* infection, IL-21 facilitates the infiltration of T cells in lung tissue, particularly CD8^+^ T cells, to effectively eliminate pathogens [26]. IL-21 activated NK cells produce IFN-γ, perforin, granzyme B, and granulosin, which effectively lyse *M. tuberculosis*-infected monocytes and inhibit the growth of *M. tuberculosis* [27]. In respiratory syncytial virus infection, IL-21 inhibits the expression of RORγt and T-bet at the site of infection, resulting in reduced recruitment of Th17 cells and Th1 cells [28]. In methicillin-resistant *S. aureus* (MRSA) infection, IL-21 has been shown to facilitate neutrophil infiltration and augment granzyme-mediated MRSA clearance [29]. During pneumonia virus of mouse (PVM) infection, IL-21R deficiency results in a decrease in the numbers of neutrophils, CD4^+^ T cells, CD8^+^ T cells and γδT cells within the pulmonary tissue, thereby leading to enhanced survival rates [30]. Our previous research revealed that IL-21/IL-21R may aggravate chlamydial lung infection by inhibiting Th1 and Th17 responses [31], as well as enhance neutrophil inflammation by regulating the TLR/MyD88 signal pathway [8]; however, whether IL-21/IL-21R modulate other immune cells remains to be studied.

In this study, combined with the suggestions from the bioinformatic analysis, we used the animal models of *C. muridarum* respiratory infection of IL-21R^−/−^ mice and the RAW264.7 macrophage cell lines to clarify the role of IL-21/IL-21R on Mφ in vivo and in vitro. Our results revealed that IL-21/IL-21R could promote the infiltration of pulmonary Mφ and induce polarization towards M1-type pro-inflammatory Mφ during *C. muridarum* respiratory infection. Adoptive transfer of M1-like bone marrow-derived macrophages (BMDM) derived from IL-21R^−/−^ mice, unlike those from WT mice, experienced lower chlamydial loads with relieved pulmonary pathology. These data indicate the negative effect of IL-21/IL-21R that occurs by modulating M1-type Mφ function to weaken the host immunity against *C. muridarum* lung infection.

## 2. Results

### 2.1. IL-21/IL-21R Promotes Pulmonary Macrophage Infiltration following C.muridarum Respiratory Infection

In order to test the participation of interleukin-21 and its receptor (IL-21/IL-21R) in macrophages (Mφ) against chlamydial infection, we quantified IL-21R expression on the pulmonary Mφ by flow cytometry. The results revealed that pulmonary Mφ (CD45^+^F4/80^+^ cells) (Figure 1A) exhibit a basal expression of IL-21R, which steadily increases to day 7 and 14 post-infection (p.i.) (Figure 1B,C). Next, we used IL-21R-deficient mice (IL-21R^−/−^) with Wild type (WT) mice as a control, to establish a *Chlamydia muridarum* (*C. muridarum*) respiratory tract infection model. The results showed that *C. muridarum* respiratory infection elicits a marked augmentation in both the proportion and absolute number of Mφ within the lung of both genotypes. On day 3 p.i., the IL-21R^−/−^ mice exhibited a significantly reduced proportion of Mφ compared with their WT counterparts. Furthermore, the absolute number of Mφ in the lungs of the IL-21R^−/−^ mice was comparatively lower than that observed in WT mice on day 7 p.i. (Figure 1D,E). These findings were further confirmed through immunofluorescence staining, which revealed a significant reduction in F4/80^+^ cell infiltration within the lungs of IL-21R^−/−^ mice compared with their WT counterparts on day 3 p.i. (Figure 1F). We conducted a simultaneous analysis of the percentages of macrophage subtypes, AMs and IMs, in the lung. It was found that, compared with WT mice, the infiltration of AMs and IMs in IL-21R^−/−^ mice was significantly reduced on day 3 p.i. and day 7 p.i., respectively (Figure 1G–I). In aggregate, these findings suggest a stimulative role of IL-21/IL-21R on Mφ infiltration against *C. muridarum* infection.

### 2.2. Transcriptome Analysis Suggests the Participation of IL-21/IL-21R in M1 Polarization and Inflammatory Responses of Macrophages during Chlamydial Respiratory Infection

Typically, polarized macrophages can be divided into classically activated macrophages (M1) and alternatively activated macrophages (M2), and by altering polarized phenotype, macrophages effectively regulate inflammatory responses [32]. Our exploratory analysis of RNA-seq datasets for the lung tissues of WT and IL-21R^−/−^ mice on day 7 following *C. muridarum* infection focused on scrutinizing Mφ transcriptome alterations to predict the impacts of IL-21/IL-21R on Mφ polarization and inflammatory responses. We firstly intersected the differentially expressed genes (DEGs) with M1- or M2-related genes in the GEO database, which revealed a significant downregulation of M1-related genes expression in the IL-21R^−/−^ group. However, the levels of M2-related genes were indeterminate (Figure 2A,B). Further, we intersected the DEGs with macrophage-related genes (MRGs) in the GEO database and obtained 153 differentially expressed genes related to Mφ (DEMRGs) (Figure 2C). Through gene ontology (GO) and Kyoto Encyclopedia of Genes and Genomes (KEGG) pathway enrichment analysis based on the DEMRGs, we clarified the potential molecular mechanisms and signaling pathways. In the functional enrichment results, we screened five items of biological processes (BP), cellular components (CC) and molecular functions (MF) in the GO analysis which were related to inflammation (e.g., leukocyte migration, regulation of inflammatory response, cytokine-mediated signaling pathway, cell chemotaxis, leukocyte chemotaxis), phagocytosis (e.g., phagocytic cup), and cytokines (e.g., cytokine receptor binding, cytokine activity, growth factor receptor binding, cytokine binding and chemokine activity) (Figure 2D). The KEGG analysis also revealed a marked induction of inflammation and cytokine pathways, including the cytokine–cytokine receptor interaction, NF-κB signaling pathway, IL-17 signaling pathway, TNF signaling pathway, Toll-like receptor signaling pathway and inflammatory bowel disease (Figure 2E). Collectively, the transcriptome analyses suggest that IL-21R deficiency may lead to a more significant downregulation of M1-related genes, and indicate a close association between IL-21/IL-21R and the inflammatory response of Mφ during *C. muridarum* infection.

### 2.3. IL-21/IL-21R Is Required for Lung Macrophage Polarization towards M1 Phenotypes during C. muridarum Respiratory Infection

Based on the results of the transcriptome analysis, we speculate that IL-21 may affect the inflammation level after chlamydia infection by regulating macrophage polarization, so we firstly explore the M1/M2 balance in the lungs of WT mice and IL-21R^−/−^ mice on days 0, 3, 7, and 14 p.i. by flow cytometry. The results revealed a higher expression of M1-type markers after infection, including consistent increases in CD80, CD86, and MHC II, whereas the percentages of CD206 (considered an M2-associated marker) exhibited a continuous decrease, indicating the M1 polarization of macrophages during *C. muridarum* infection. On day 3 or day 7 p.i., a significant reduction in CD80^+^, CD86^+^ and MHC II^+^ cells was observed for total Mφ, AMs and IMs of IL-21R^−/−^ mice compared with their WT counterparts (Figure 3A–F). It is worth noting that the percentages of M2 (F4/80^+^CD206^+^) macrophages in total Mφ, AMs and IMs of IL-21R^−/−^ mice were, unexpectedly, no different from those in WT mice (Figure 3G,H). These findings suggest that during *C. muridarum* respiratory infection, IL-21/IL-21R might promote the expression of M1-related cell surface markers on pulmonary Mφ.

### 2.4. IL-21/IL-21R Promotes Pro-Inflammatory Cytokine Production from Pulmonary Macrophages during C.muridarum Respiratory Infection

One of the most crucial attributes of Mφ responses lies in their ability to secrete cytokines, so we further analyzed how IL-21/IL-21R influence the Mφ polarization after infection. Firstly, we detected the pro-inflammatory (TNF-α) and anti-inflammatory cytokine (including TGF-β and IL-10) secretion from the lung total Mφ, AMs, and IMs in vivo through flow cytometry. As shown in Figure 4A–F, lung total Mφ and IMs of IL-21R^−/−^ mice produced significantly less TNF-α on day 7 p.i. compared with those of their WT counterparts, while the AMs of IL-21R^−/−^ mice exhibited significantly less TNF-α production on day 3 p.i. There was no significant difference in TGF-β or IL-10 secretion between two groups among lung Mφ, AMs, and IMs. In vitro, the pro-inflammatory (including TNF-α and IL-12 p40) and anti-inflammatory (including TGF-β and IL-10) cytokine levels in RAW264.7 cells stimulated by *C. muridarum* were detected, with LPS-induced M1 polarization used as a positive control. *C.muridarum* infection significantly upregulated the mRNA expression of the pro-inflammatory cytokines TNF-α and IL-12p40 in the RAW264.7 cells, and rmIL-21 addition further promoted the pro-inflammatory cytokine levels of RAW264.7 cells (Figure 4G,H and Table 1). As shown in Figure 4I,J, no significant alteration was observed in the mRNA expression of IL-10 and TGF-β when rmIL-21 were added. These findings reveal that IL-21/IL-21R promotes the pro-inflammatory cytokine production of pulmonary macrophages, further suggesting the M1 polarization of *C. muridarum*-stimulated Mφ.

### 2.5. IL-21/IL-21R Could Enhance the Function of M1-Type Macrophages during C.muridarum Infection

Nitric oxide (NO), an M1-activation marker, is a crucial pro-inflammatory mediator in the pathogenesis of several inflammatory disease [33,34]. The release of NO is tightly regulated by the iNOS level, whose expression depends on reactive oxygen species (ROS) accumulation [34,35]. ARG1, involved in the conversion of L-arginine to L-ornithine and polyamines, is reported to be an important marker of M2 function [22]. Given that IL-21 significantly enhances the M1-type polarization of Mφ stimulated with *C. muridarum*, both in vivo and in vitro, we further investigated the levels of function markers of M1 and M2. As depicted in Figure 5A,B, the production of iNOS by lung total Mφ, AMs, and IMs in both genotypes exhibited a continuous increase, and the production of ARG1 showed a continuous decrease following *C. muridarum* infection. Lung total Mφ, AMs and IMs from the IL-21R^−/−^ mice produced significantly less iNOS compared with those from WT mice on day 3, 7, and 14 p.i., whereas there was no significant difference in the ARG1 production between WT and IL-21R^−/−^ mice among total Mφ, AMs and IMs (Figure 5C,D).

Subsequently, we stimulated the RAW264.7 cells with *C. muridarum* in vitro and quantified the NO production in the culture supernatants from different groups at 3 h, 6 h, 12 h, and 24 h post-infection. NO levels were detected in the supernatants from *C. muridarum*-stimulated RAW264.7 cells at 12 h and 24 h, and rmIL-21 treatment further induced NO production at 24 h (Figure 5E). RAW264.7 cells from each group were collected at 24 h after infection and analyzed for iNOS and ARG1 mRNA expression using qPCR. Consistent with the results obtained for NO production, mRNA expression of iNOS significantly increased following stimulation by *C. muridarum* alone, and rmIL-21 addition further enhanced iNOS expression, whereas there was no significant difference in ARG1 mRNA expression (Figure 5F and Table 1). Although the ROS response exhibited a robust fluorescence in the *C. muridarum* group, there was no significant difference with the addition of rmIL-21 (Figure 5G). Further, the effects of rmIL-21 were evaluated in vivo. The C57BL/6 mice were administered with rmIL-21 or an equal volume of PBS prior to being challenged with *C. muridarum*. Elevated levels of iNOS were observed in pulmonary total Mφ, AMs, and IMs of rmIL-21-administered mice compared with PBS-administered mice (Figure 5H,I). Collectively, these results further support the role of IL-21/IL-21R signaling in promoting M1 function during *C. muridarum* infection.

### 2.6. IL-21/IL-21R Might Aggravate M1-Mediated Pulmonary Inflammation following C. muridarum Infection

To directly investigate the impact of IL-21/IL-21R on M1-mediated inflammatory effects, we compared the immunopathological consequences of recipient mice adoptively transferred with M1-like BMDMs derived from WT (WT-BMDM-M1) or IL-21R^−/−^ (IL-21R^−/−^-BMDM-M1) mice. As illustrated in the Appendix A, M1-like BMDM were induced in vitro and their purity was validated. The C57BL/6 mice received PBS, WT-BMDM-M1, or IL-21R^−/−^-BMDM-M1 via the tail vein and then were infected intranasally with *C. muridarum* 2 h later. The body weight loss, pulmonary bacterial growth, pulmonary pathology, and cytokine profiles in the recipients were analyzed after infection. Mice that received either WT-BMDM-M1 or IL-21R^−/−^-BMDM-M1 showed similar changes in body weight loss during *C. muridarum* infection (Figure 6A). Significantly higher levels of inclusion forming units (IFUs) and severe pathological changes appeared in the lungs of WT-BMDM-M1 recipients compared with the PBS recipients (Figure 6B–D), indicating the pro-inflammatory effects of M1 during *C. muridarum* infection. Compared with the WT-BMDM-M1 recipients, the IL-21R^−/−^-BMDM-M1 recipients exhibited a reduced bacterial burden (Figure 6B), moderate inflammatory pathology (Figure 6C) and lower inflammatory grades (Figure 6D) on day 14 p.i. Consistent with disease conditions, analysis of inflammatory cytokine expression in the lungs showed that mice receiving IL-21R^−/−^-BMDM-M1 exhibited lower levels of IL-1β, IL-6 and TNF-α compared with those receiving WT-BMDM-M1 (Figure 6E and Table 1). These findings reveal that IL-21/IL-21R can promote the pro-inflammatory effects of M1 after *C. muridarum* infection.

## 3. Materials and Methods

### 3.1. Mice

Female C57BL/6 mice (wild-type, WT) of 6–8 weeks were purchased from Huafukang Biotechnology (Beijing, China). IL-21R^−/−^ mice on a C57BL/6 background were obtained from Prof. Zhinan Yin (Nankai University, Tianjin, China). The mice were housed in pathogen-free conditions under a standardized light–dark cycle with free access to food and water at Tianjin Medical University. All the animal experiments were approved by the Committee on the Ethics of Animal Experiments of Tianjin Medical University.

### 3.2. C.muridarum Respiratory Tract Infection Models

*Chlamydia muridarum* (*C. muridarum*), obtained from Dr. Xi Yang (the University of Manitoba, Canada), was cultured, purified, and enumerated as previously described [8,36]. The mice were anesthetized with isoflurane using an anesthesia machine and intranasally inoculated with 1 × 10^3^ inclusion forming units (IFUs) of *C. muridarum* in 40 μL of sucrose phosphate glutamic acid (SPG) buffer.

### 3.3. Lung Single Cells Preparation

Lung single cells of mice were prepared and analyzed as described previously [6,8,37]. Briefly, sterile isolations of infected lung at different time points were digested in RPMI 1640 containing 2 mg/mL collagenase XI (Sigma-Aldrich, St. Louis, MO, USA) for 55 min at 37 °C, 5% CO_2_, and 2 mM EDTA was added at the last 5 min of incubation. After enzyme digestion, 35% Percoll (GE Healthcare, Chicago, London, England) was added and centrifuged at 12 °C, 2000 rpm for 20 min to remove tissue fascia. ACK Lysis buffer (Tris-NH_4_Cl) was used to lyse the erythrocytes. The single cells were resuspended in complete RPMI-1640 medium (RPMI-1640 supplemented with 10% heat inactivated fetal bovine serum (FBS, Shanghai Life iLab Biotech, Shanghai, China), 0.05 mmol/L 2-mercaptoethanol, 100 U/mL penicillin, and 0.1 mg/mL Streptomycin (Solarbio, Beijing, China) for subsequent experiments.

### 3.4. RAW 264.7 Culture and Treatment

The RAW264.7 macrophages cell line was obtained from the American Type Culture Collection (ATCC). The RAW264.7 cells were grown in 24-well plates at a density of 5–6 × 10^5^ cells per well in DMEM (Gibco, CA, USA) medium with 10% FBS and 1% penicillin-streptomycin antibiotics. The cells were treated with 5–6 × 10^6^ IFUs of UV-inactivated *C. muridarum* (multiplicity of infection (MOI = 10)) for 2 h, and then added 100 ng/mL recombinant murine IL-21 (rmIL-21, PEPROTECH, 5 Cedarbrook Drive, Cranbury, NJ, USA) or 100 ng/mL LPS (Sigma-Aldrich, St. Louis, MO, USA) (which served as positive controls) for 3 h, 6 h, 12 h, and 24 h. Cells were harvested for mRNA expression and ROS activity assays, while the cell culture supernatants were harvested for NO production detection.

### 3.5. Flow Cytometry

For cell-surface staining, lung single cells were incubated with CD16/CD32 (Invitrogen, Carlsbad, CA, USA) for 30 min in the dark at 4 °C to block non-specific Fc staining followed by washing with PBS containing 2% FBS. The samples were divided and stained using anti-CD45-PerCP, anti-F4/80-APC, anti-CD11c-FITC, anti-CD80-PE-Cy7, anti-CD86-PE, anti-MHC II-PE, anti-CD206-PE-Cy7, and anti-IL-21R-PE (all purchased from Biolegend, San Diego, CA, USA) for 30 min at 4 °C in the dark to stain the macrophages. For intracellular cytokine staining, lung single cells were stimulated with PMA (50 ng/mL, Sigma-Aldrich), ionomycine (1μg/mL, MCE, Monmouth Junction, NJ, USA), and brefeldin A (10 μg/mL, BioLegend, San Diego, CA, USA) for 5–6 h at 37 °C. Fc segments were blocked with anti-CD16/32 and stained first for surface antigens (CD45-PerCP, F4/80-APC, and anti-CD11c-FITC). Then, the cells were fixed with Fixation Buffer (Biolegend) for 20 min in the dark at room temperature, permeabilized with 1 × Intracellular Staining Perm Wash Buffer (Biolegend) for 30 min at room temperature, and subsequently incubated in the dark with anti-TNFα-PE, anti-TGFβ-PE-Cy7, and anti-IL-10-PE antibodies for 40 min at room temperature. For iNOS and ARG1 staining, the cells were stained for surface antigens, fixed, permeabilized with 1 × Intracellular Staining Perm Wash Buffer, and stained for iNOS-PE and ARG1-PE-Cy7 for 40 min at room temperature. Compensation for spectral mixing was applied by flow cytometric analysis of cells singly stained with antibodies against CD4 or CD3 with PeCP, APC, PE-Cy7, FITC and PE. Flow cytometry was performed using a FACS Canto II flow cytometer (BD Biosciences, Franklin, NJ, USA), and the data were analyzed using FlowJo version 10.

### 3.6. Immunofluorescence Staining

Lung tissues, aseptically isolated on day 3 p.i., were fixed in 4% paraformaldehyde (PFA) (Shanghai Life iLab Biotech) for 24 h at 4 °C. Subsequently, the tissues were subjected to gradient dehydration using 15% and 30% sucrose solutions for another 24 h before being embedded in Tissue-Tek O.C.T. Compound (SAKURA, Baltimore, MD, USA) and sectioned into cryosections with a thickness of 8 μm. The lung sections were blocked with 10% goat serum for 1 h at room temperature and incubated with anti-F4/80 antibody (1:150 dilution, ab16911, abcam, Cambridge, England) overnight at 4 °C. Next, they were rewarmed for 1 h before being incubated with Goat Anti-Rat IgG H&L (Alexa Fluor 488) preadsorbed secondary antibody (1:300 dilution, ab150165, abcam) for 1 h at room temperature in the dark. This was followed by washing five times using PBS. Then, AutoFluoQuencher (APPLYGEN, Beijing, China) was added and incubated for 15 min. Slides were then sealed by adding DAPI Fluoromount-G (SouthernBiotech, UAB, Birmingham, AL, USA), and images were acquired using a fluorescent microscope (200×).

### 3.7. Quantitative Real-Time PCR(qPCR)

At the time of harvest, the cells were washed with sterile PBS 2–3 times, and TRIzol Reagent (Ambion, Austin, Texas, USA) was used to extract the total RNA from the cells in accordance with the manufacturer’s protocol, which was then reverse transcribed into cDNA using the TransScript One-Step gDNA Removal and cDNA Synthesis SuperMix (Transgen, Beijing, China). Real-time quantitative PCR (qPCR) was performed using the 2 × RealStar Green Fast Mixture (GenStar, Beijing, China) using a Light Cycler 96 (Roche, Basel, Switzerland), and the expression levels of the target genes were measured through absolute quantification. β-Actin was used as an endogenous reference. The primers were purchased from Shanghai Sangon Biotech. The primers used are shown in Table 1. 

### 3.8. NO Analysis

Nitrite concentrations in the culture media were quantified using a commercial Nitric Oxide Assay Kit (Beyotime, Shanghai, China) according to the manufacturer’s instructions. Briefly, cell culture supernatants obtained at different time points were incubated with Griess Reagent I and Griess Reagent II at room temperature for either 10 or 30 min. The absorbance was measured at 540 nm and nitrite levels were calculated using a standard curve of sodium nitrite.

### 3.9. ROS Analysis

ROS production in the infected RAW264.7 cells was measured using a commercial ROS assay kit (Beyotime) according to the manufacturer’s instructions. Briefly, the culture medium was removed and 200 μL of 2,7-Dichlorodihydrofluorescein diacetate (DCFH-DA) diluted with DMEM (1:1000) was added, which was followed by incubation for 20 min at 37 °C. Following three washes with DMEM, the cells were collected and analyzed by flow cytometry. The fluorescence intensity of FITC reflects the production of ROS.

### 3.10. Administration of rmIL-21

For administration of rmIL-21, the C57BL/6 mice were inoculated intranasally with 0.5 μg rmIL-21 (PEPROTECH, 5 Cedarbrook Drive, Cranbury, NJ, USA) in 20 μL PBS at 1 day before infection and at days 0, 2, 4, and 6, and the control group was given 20 μL sterile PBS on the same schedule. The mice were euthanized at the designated time points after infection. Lung tissue was isolated and lung single cell was prepared for flow cytometry to detect iNOS.

### 3.11. Induction and Adoptive Transfer of M1-Type Bone Marrow-Derived Macrophages (BMDM-M1)

Bone marrow (BM) cells were flushed from the femur and tibia of WT or IL-21R^−/−^ mice at similar ages (6–8 weeks) with DMEM supplemented with 1% penicillin and streptomycin. ACK Lysis buffer (Tris-NH_4_Cl) was used to lyse erythrocytes. The BM cells were cultured in DMEM supplemented with 10% FBS, 1% penicillin and streptomycin, and 20 ng/mL recombinant macrophage colony-stimulating factor (M-CSF, PEPROTECH) for 7 days at 37 °C in 6-well tissue culture plates. The medium was changed on days 3, 5, and 7. On day 7, the cells were treated with DMEM complete medium containing 100 ng/mL IFNγ for 24 h to induce M1 polarization. A total of 1 × 10^6^ BMDM-M1 cells in 200 μL of PBS were adoptively transferred into naïve C57BL/6 mice via tail intravenous injection, followed by *C. muridarum* infection at 2 h post-transfer.

### 3.12. Pulmonary Chlamydia Loads and Histopathological Analysis

Mice were intranasally infected with *C. muridarum* and were killed at the indicated time points. Lung tissues were homogenized in SPG buffer. To determine chlamydial growth in vivo, lung homogenates were titrated by infection of HeLa cell monolayers as described previously [8,37]. For histopathological analysis, the aseptically separated lungs were fixed in 4% PFA for 48 h. The lung sections were stained with H&E according to the manufacturer’s instructions (Solarbio), and the histological changes were analyzed by light microscopy. The severity of lung inflammation and injury was scored as described previously [37,38].

### 3.13. Bioinformatic Analysis

Isolated lung tissues of WT and IL-21R^−/−^ mice on day 7 after *C. muridarum* respiratory tract infection were sequenced by Biomarker Technologies Co., Ltd. (Beijing, China). The obtained data were normalized for preprocessing, and differentially expressed genes between WT and IL-21R^−/−^ were screened via R language “limma” package. Genes with |log2 Fold changes| > 2 and p < 0.05 were considered as differentially expressed genes (DEGs). Mφ-related genes (MRGs) were searched in the GEO database (http://www.ncbi.nlm.nih.gov/geo, online database, accessed on 3 January 2023) and overlapping genes between DEGs and MRGs were defined as differentially expressed Mφ-related genes (DEMRGs). Then, the Gene Ontology (GO) enrichment and Kyoto Encyclopedia of Genes and Genomes (KEGG) pathway analysis of DEMRGs was carried out using the “clusterProfiler” package of R software. The GO analysis included three categories: biological process (BP), cellular component (CC), and molecular function (MF). This analysis was performed to evaluate the significance of the terms compared with two genotypes of mice. KEGG analysis was used to explore potential pathways.

### 3.14. Statistical Analysis

The data were analyzed using GraphPad Prism 8 (GraphPad InStat Software, San Diego, CA, USA). Differences between two different groups were assessed using two-way ANOVA followed by Šidák’s multiple comparisons test. Differences among multiple groups were analyzed by one-way ANOVA followed by Dunnett’s multiple comparisons test. Results with p values of 0.05 or less were considered to be statistically significant different.

## 4. Discussion

In our previous investigations, we observed an upregulation in the expression of both IL-21 and IL-21R within lung tissue from infected C57BL/6 mice [31]. Here, we further observed increased expression of IL-21R on pulmonary Mφ. Previous studies demonstrated that IL-21R^−/−^ mice exhibit resistance to *C. muridarum* infection and display significant accumulation of Th1 and Th17 cells on day 7 p.i., while administration of rmIL-21 exacerbates lung infection in C57BL/6 mice [31]. We also found that IL-21/IL-21R may exacerbate neutrophilic inflammation by modulating the TLR/MyD88 signaling pathway during *C. muridarum* respiratory tract infection [8]. Despite studies indicating that neutrophil recruitment is preceded by a period of resistance to macrophage-mediated clearance in hypervirulent Klebsiella pneumoniae infection [27]. We observed distinctly attenuated infiltration of Mφ in the lungs of IL-21R^−/−^ mice during the early stage of *C. muridarum* respiratory tract infection in this study.

The regulation of IL-21 on Mφ polarization has been reported in many fields. In pulmonary arterial hypertension, IL-21 promoted the polarization of primary alveolar macrophages toward the M2 phenotype [22]. In rheumatoid arthritis, IL-21 impairs pro-inflammatory activity of M1-like Mφ, but promotes M2 polarization of Mφ [39]. Tumor local delivery of IL-21 can skew TAM polarization away from the M2 phenotype to a tumor-inhibiting M1 phenotype, which rapidly stimulates T cell responses against tumors [40]. In Alzheimer’s Disease, increased IL-21 levels lead to increased expression of MHC II in spleen Mφ and secretion of a large number of pro-inflammatory cytokines such as IL-18, TNFα and IL-1β [41]. In the current study, transcriptome analysis showed that M1-related genes and a majority of M2-related genes were significantly downregulated in IL-21R^−/−^ mice on day 7 post *C. muridarum* respiratory infection. Our in vivo and in vitro findings provide further compelling evidence that the IL-21/IL-21R signaling pathway plays a crucial role in M1 polarization and the pro-inflammatory effects of lung macrophages during *C. muridarum* respiratory infection, while it has no significant impacts on the M2 phenotype.

Our study revealed a significant reduction in the secretion of the pro-inflammatory mediators TNF-α and iNOS in lung macrophages of IL-21R^−/−^ mice, while there was no significant alteration observed in the secretion of the anti-inflammatory mediators TGFβ, IL-10 and ARG1. This was demonstrated through rmIL-21 and *C. muridarum* stimulation of the RAW264.7 cell line in vitro. Upon addition of rmIL-21 to C. muridarum-stimulated RAW264.7 cells, a significant upregulation in the expression of the pro-inflammatory cytokines IL-12p40 and TNF-α, as well as the pro-inflammatory mediators iNOS and NO, was observed, whereas no significant differences were noted in the expressions of the anti-inflammatory mediators TGFβ, IL-10 and ARG1. Therefore, we believe that the IL-21/IL-21R signaling pathway is involved in exacerbating the pro-inflammatory response of C. muridarum-stimulated macrophages. In the context of ROS, it is possible that IL-21 may not be induced through this pathway during the process of promoting Mφ-mediated secretion of iNOS and NO. Further investigation is necessary to elucidate the underlying mechanism.

The participation of IL-21/IL-21R in chemotaxis was revealed in many previous studies. It was demonstrated that IL-21R is essential for pancreas DCs to acquire the CCR7 chemokine receptor and migrate into the draining lymph node, thereby breaking diabetes resistance in type 1 diabetes [42]. Stimulation of intestinal epithelial cells with IL-21 resulted in an upregulation of macrophage inflammatory protein-3 alpha (MIP-3alpha), a T-cell chemoattractant that enhances lymphocyte migration [43]. Here, as revealed in the RNA-seq results, it is noteworthy that many DEMRGs are related to chemotaxis. It is emphasized that the IL-21/IL-21R signaling may play a role in recruiting mononuclear macrophages from the peripheral blood to infected sites during *C. muridarum* respiratory infection. We believe that the IL-21/IL-21R-regulated macrophage chemotaxis provides a promising direction for further mechanistic exploration.

As demonstrated in Appendix A, we also found that IL-21 exerts different functional effects on M0 and M1-type Mφ cells when stimulated with C. muridarum. IL-21/IL-21R signaling is involved in inhibiting iNOS mRNA expression but promotes ARG1 mRNA expression in C. muridarum-stimulated M1-like Mφ cells. This effect of IL-21 on *C. muridarum* stimulated-M1-like Mφ has not been reported before, and its mechanism is also worth further investigation.

IL-21/IL-21R signaling is involved in regulating diseases caused by various pathogens, not just chlamydia, and chlamydial respiratory infection is not solely associated with this signal. Our previous research revealed that IL-27 signaling can support the Th1 response by inhibiting IL-10 production in DCs, suppressing excessive Th17 responses, and reducing neutrophil inflammation, which mediates protective host defenses against chlamydial respiratory infection in mice [37,44]. In vivo evidence suggests that IL-17/Th17 can promote Th1 immunity by modulating DC function to defend against chlamydial respiratory infection [6]. We also observed that FcγRI may regulate host immunity and the inflammatory response during chlamydial infection since the Th1 response was enhanced while the pro-inflammatory M1 macrophage polarization was limited in Fcgr1^−/−^ mice [38]. Although these studies are mainly based on animal models, there may be some differences with the immune response of human chlamydia infection; however, the results of these studies provide new ideas and directions for developing effective anti-chlamydia drugs or vaccines.

## 5. Conclusions

In summary, our research highlights the critical role of IL-21/IL-21R signaling in promoting macrophage polarization towards M1 phenotypes and the pro-inflammatory effects of M1 during *C. muridarum* respiratory infection. And we have confirmed the negative effect of IL-21/IL-21R signaling that occurs by modulating M1 function to weaken the host immunity against *C. muridarum* lung infection, which may have implications in developing effective chlamydial vaccines and in the understanding of host defense mechanisms in other lung infections.

## Figures and Tables

**Figure 1 ijms-24-12557-f001:**
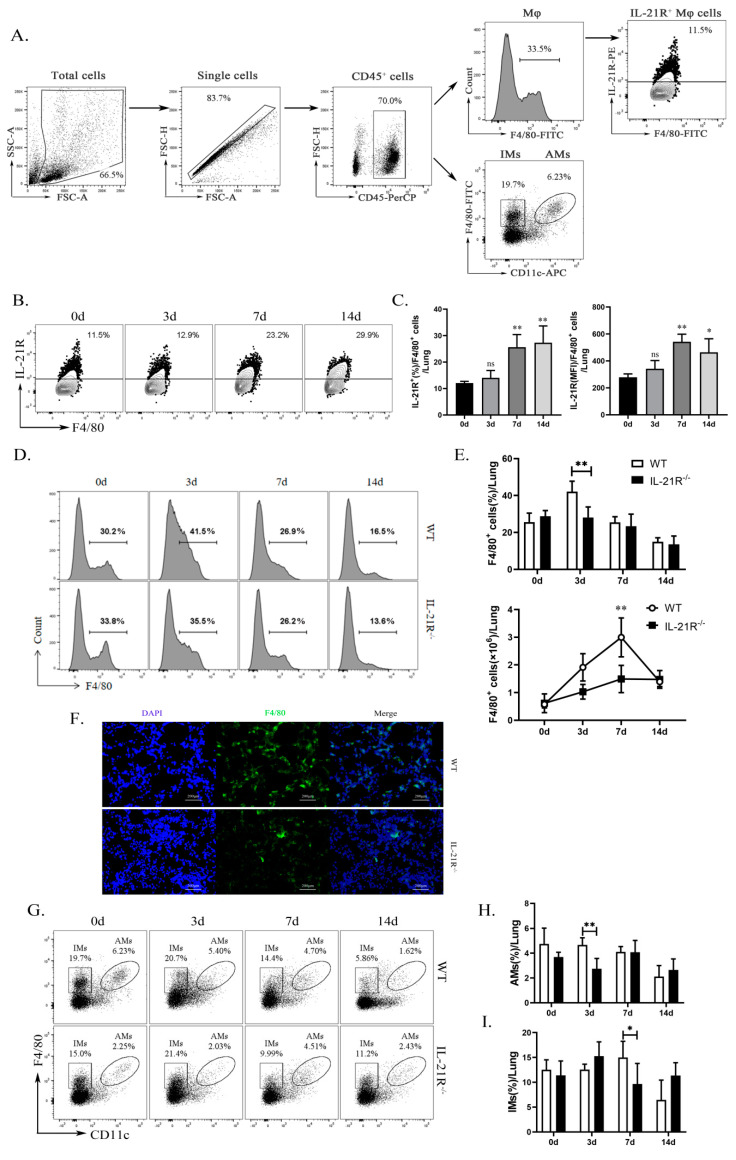
IL-21R^−/−^ mice exhibited reduced pulmonary macrophages (Mφ) infiltration during respiratory infection with *C. muridarum*. WT and IL-21R^−/−^ mice were infected with 1 × 10^3^ inclusion forming units (IFUs) of Chlamydia muridarum (*C. muridarum*) via the respiratory tract. Lung tissue was retrieved from infected animals on days 0, 3, 7, and 14 post infection (p.i.) and processed into a single-cell suspension. Flow cytometry was employed to scrutinize the populations of CD45^+^F4/80^+^ Mφ, CD45^+^F4/80^+^CD11c^hi^ AMs and CD45^+^F4/80^+^CD11c^−/lo^ IMs in lung. (**A**) Gating strategy for lung Mφ, AMs, IMs and IL-21R^+^ Mφ. (**B**) Representative flow cytometry images of IL-21R^+^ Mφ in lung tissues from C57BL/6 mice on days 0, 3, 7, and 14 p.i. (**C**) The percentages of IL-21R^+^ Mφ and the mean fluorescence intensity (MFI) of IL-21R expression on lung Mφ. (**D**) Representative flow cytometry images of CD45^+^F4/80^+^ Mφ in lung tissues from WT and IL-21R^−/−^ mice on different days. (**E**) The percentages (up) and the absolute numbers (down) of CD45^+^F4/80^+^ Mφ in lung tissues from WT and IL-21R^−/−^ mice on different days. (**F**) Representative immunofluorescence images of frozen lung sections from WT and IL-21R^−/−^ mice on day 3 p.i. (Green, F4/80^+^ cells; Blue, DAPI) (200×). (**G**) Representative images showing flow cytometry analysis of AMs and IMs in the lungs of WT and IL-21R^−/−^ mice on different days. (**H**,**I**) The percentages of AMs (**H**) and IMs (**I**) in lung tissues from WT and IL-21R^−/−^ mice on different days. Data are presented as means ± SD from *n* = 3–4 animals per genotype and time point. Data show one representative of three independent experiments. Statistical significances of differences were determined by one-way ANOVA (**C**) and two-way ANOVA (**E**,**H**,**I**). * *p* < 0.05, ** *p* < 0.01. “ns” indicates not significant.

**Figure 2 ijms-24-12557-f002:**
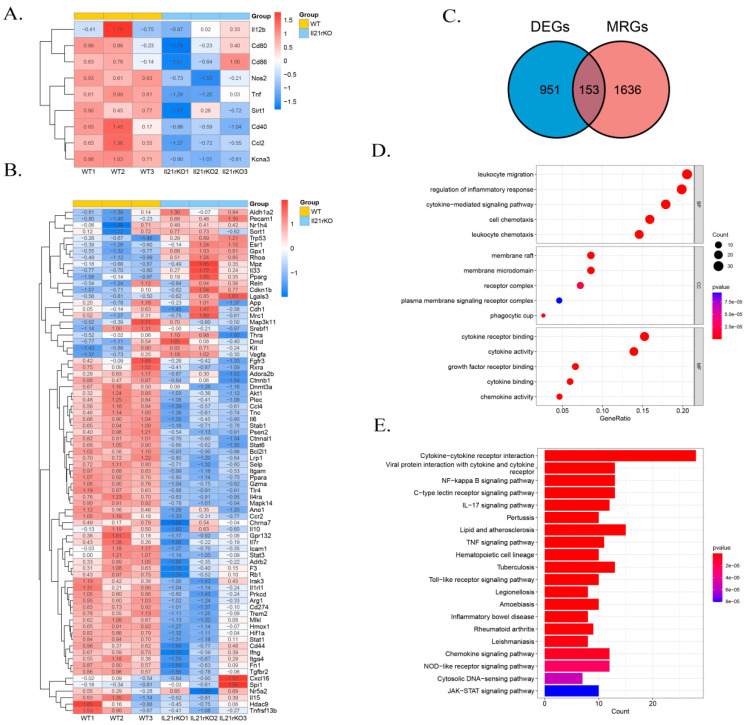
IL-21/IL-21R are involved in the M1 polarization and inflammatory responses of Mφ during *C. muridarum* respiratory infection as determined by RNA sequencing (RNA-seq). RNA-seq analysis of gene expression profile was performed on lung tissues extracted from WT (*n* = 3) and il21r knockout (il21r KO) (*n* = 3) mice on day 7 of *C. muridarum* respiratory infection. (**A**,**B**) Heatmap displaying the relative expression levels of M1-related genes (**A**) and M2-related genes (**B**) in WT and il21r KO mice. Gene expression values are indicated by the color intensities of red (upregulated) or blue (downregulated). (**C**) Venn diagram showing the overlap between the gene set of differentially expressed genes (DEGs) and the gene set macrophage-related genes (MRGs). The numbers shown in the diagram represent the number of genes in each group. (**D**) GO analysis using differentially expressed genes related to Mφ (DEMRGs) showing the top 15 GO terms. (**E**) KEGG pathway analysis using DEMRGs showing the top 20 enrichment pathways. Data are presented as means ± SD from *n* = 3 animals per group.

**Figure 3 ijms-24-12557-f003:**
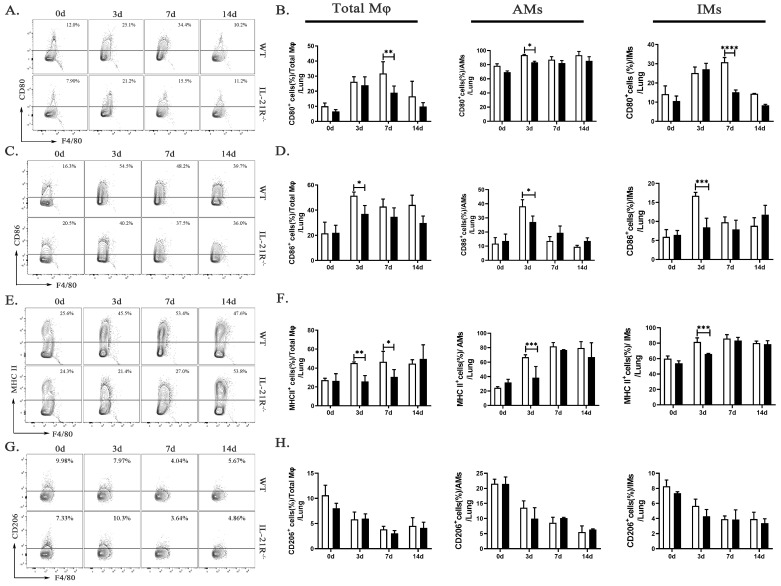
IL-21R^−/−^ mice lung M1-type Mφ significantly attenuated after *C. muridarum* respiratory tract infection. (**A**,**C**,**E**,**G**) Representative flow cytometry images of CD80^+^ (**A**), CD86^+^ (**C**), MHC II^+^ (**E**), and CD206^+^ (**G**) cells in CD45^+^F4/80^+^ Mφ from lung tissues of WT and IL-21R^−/−^ mice infected with *C. muridarum* respiratory tract on different days. (**B**,**D**,**F**,**H**) Flow cytometry data are summarized to show the percentages of CD80^+^ (**B**), CD86^+^ (**D**), MHC II^+^ (**F**), and CD206^+^ (**H**) cells within total Mφ, AMs and IMs from lung tissues of WT and IL-21R^−/−^ mice infected with *C. muridarum* respiratory tract on different days. Data are presented as means ± SD from *n* = 3–4 animals per genotype and time point. Data show one representative of three independent experiments. Statistical significances of differences were determined by two-way ANOVA. * *p* < 0.05, ** *p* < 0.01, *** *p* < 0.001, **** *p* < 0.0001.

**Figure 4 ijms-24-12557-f004:**
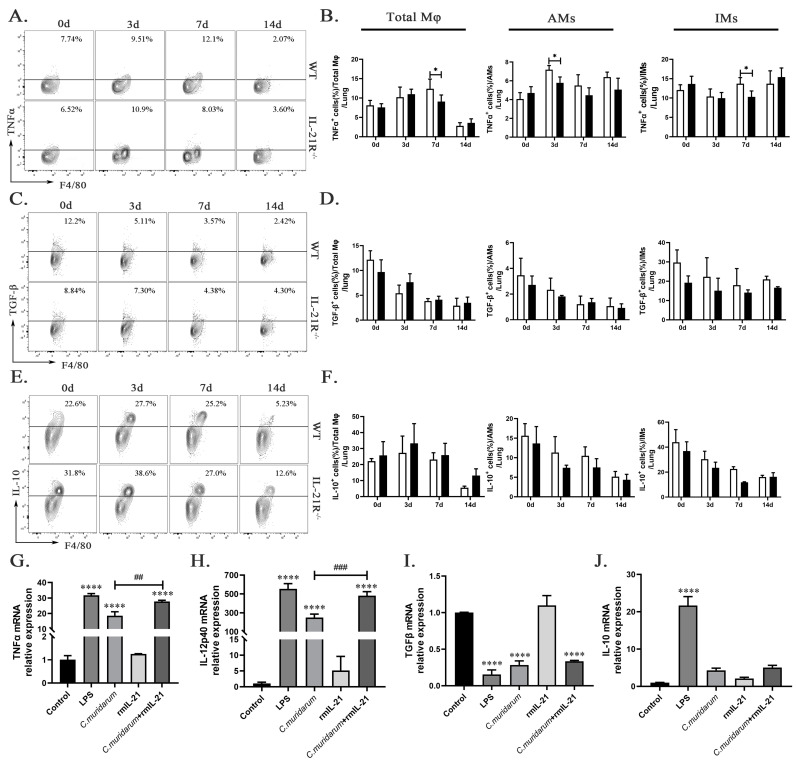
IL-21 induced Mφ to secrete pro-inflammatory cytokines in response to *C. muridarum* stimulation. (**A**,**C**,**E**) Representative flow cytometry images of TNFα^+^ (**A**), TGFβ^+^ (**C**) and IL-10^+^ (**E**) cells in total Mφ from lung tissues of WT and IL-21R^−/−^ mice infected with *C. muridarum* on different days. (**B**,**D**,**F**) Flow cytometry data are summarized to show the percentages of TNFα^+^ (**B**), TGFβ^+^ (**D**) and IL-10^+^ (**F**) cells within total Mφ, AMs and IMs from lung tissues of WT and IL-21R^−/−^ mice infected with *C. muridarum* on different days. (**G**-**J**) The mRNA expression levels of TNFα (**G**), IL-12p40 (**H**), TGFβ (**I**), and IL-10 (**J**) were quantified by quantitative real-time PCR (qPCR) in RAW264.7 cells under various conditions, including the control (no stim), LPS (100 ng/mL), *C.muridarum* (MOI = 10), recombinant mouse IL-21 (rmIL-21) (100 ng/mL), and co-treatment with *C.muridarum* (MOI = 10) and rmIL-21 (100 ng/mL). Data are presented as means ± SD from *n* = 3–4 animals per group or three multiple wells in each group. The data shown are one representative of three independent experiments. Statistical significances of differences were determined by two-way ANOVA (**B**,**D**,**F**) and one-way ANOVA (**G**-**J**). * *p* < 0.05, **** *p* < 0.0001, ## *p* < 0.01, ### *p* < 0.001. “*” indicates a comparison within a group or with the control group. “#” refers to a comparison between the *C. muridarum* group and *C. muridarum*+ rmIL-21 group.

**Figure 5 ijms-24-12557-f005:**
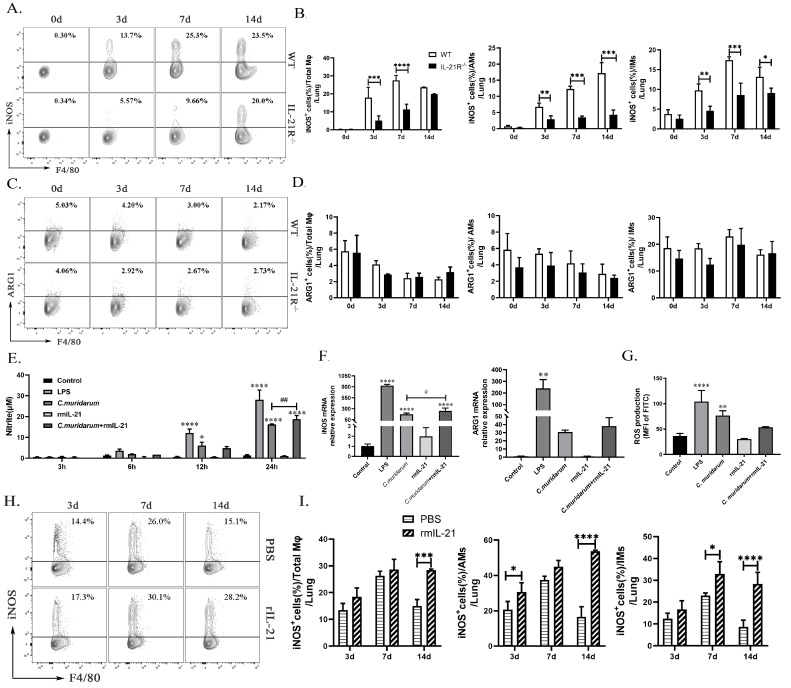
IL-21/IL-21R enhances the pro-inflammatory function of macrophages during *C. muridarum* infection. (**A**,**C**) Representative flow cytometry images of iNOS^+^ cells (**A**) and ARG1^+^ cells (**C**) in total Mφ from lung tissues of WT and IL-21R^−/−^ mice infected with *C. muridarum* on different days. (**B**,**D**) Flow cytometry data are summarized to show the percentages of iNOS^+^ cells (**B**) and ARG1^+^ cells (**D**) within total Mφ, AMs and IMs from lung tissues of WT and IL-21R^−/−^ mice infected with *C. muridarum* on different days. (**E**) Nitrite levels were quantified by Griess assay at different times of RAW264.7 treatment. (**F**) The mRNA expressions of iNOS and ARG1 were analyzed by qPCR after 24 h of treatment with RAW264.7. (**G**) Reactive oxygen species (ROS) production was analyzed by flow cytometry after 24 h of treatment with RAW264.7. (**H**) Representative flow cytometry images of iNOS^+^ cells in total Mφ from lung tissues of PBS and rmIL-21 mice infected with *C. muridarum* on different days. (**I**) Flow cytometry data are summarized to show the percentages of iNOS^+^ cells within total Mφ, AMs and IMs from lung tissue samples of PBS and rmIL-21 mice infected with *C. muridarum* on different days. Data are presented as means ± SD from *n* = 3-4 animals per group or three multiple wells for each group. The data shown are one representative of three independent experiments. Statistical significances of differences were determined by two-way ANOVA (B, D and I) and one-way ANOVA (**E**–**G**). * *p* < 0.05, ** *p* < 0.01, *** *p* < 0.001, **** *p* < 0.0001, # *p* < 0.05, ## *p* < 0.01. “*” indicates a comparison within a group or with the control group. “#” refers to the comparison between the *C. muridarum* group and *C. muridarum*+ rmIL-21 group.

**Figure 6 ijms-24-12557-f006:**
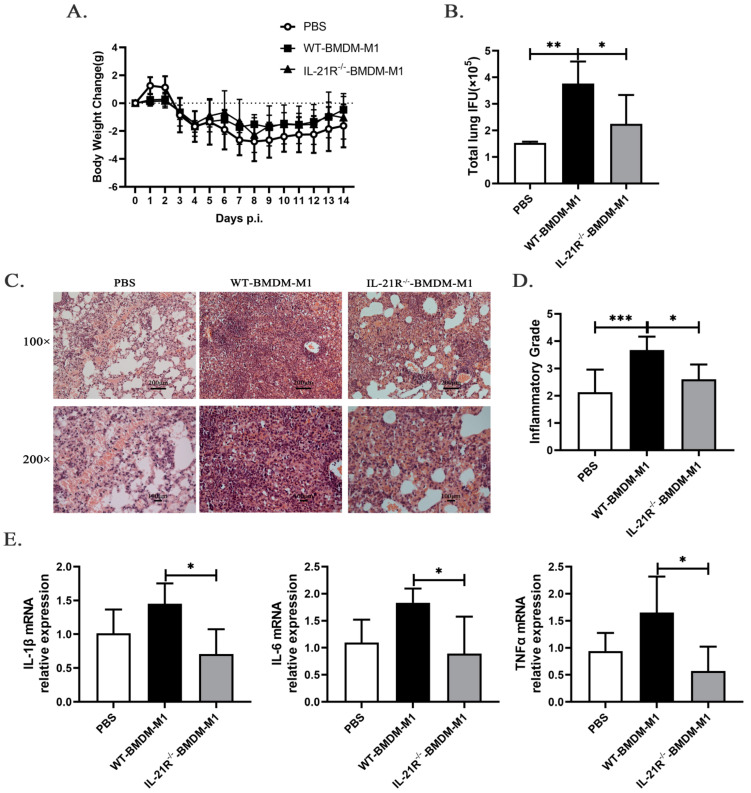
Adoptive transfer of M1-like bone marrow-derived macrophages (BMDM) from IL-21R^−/−^ mice into C57BL/6 mice ameliorates *C. muridarum*-infected lung inflammatory pathology compared with the transfer of M1-like BMDM from WT mice. Bone marrow was isolated from both the WT and IL-21R^−/−^ mice and induced in vitro to differentiate into M1-like BMDM. Subsequently, C57BL/6 mice were intravenously injected with either WT or IL-21R^−/−^ M1-like BMDM (1 × 10^6^ cells / 200μL PBS), while the control group received an equal volume of PBS. Two hours later, the recipient mice were intranasally inoculated with 1 × 10^3^ IFUs of *C. muridarum*. (**A**) Body weight changes were monitored for 14 days after infection in different groups. (**B**) The chlamydial burdens in the lung at day 14 p.i. were determined by quantifying the levels of IFUs. (**C**) Representative H&E staining images of lungs in each group were observed under light microscopy at magnifications of ×100 (up) and ×200 (down). (**D**) Inflammatory grades were assessed based on the H&E staining results as described in the Materials and Methods. (**E**) The mRNA expression levels of IL-1β, IL-6, and TNFα were quantified by qPCR in lung tissues from the different groups. The data are presented as means ± SD from *n* = 3–4 animals per group. The data shown are one representative of three independent experiments. Statistical significances of differences were determined by one-way ANOVA. * *p* < 0.05, ** *p* < 0.01, *** *p* < 0.001.

**Table 1 ijms-24-12557-t001:** Primer sequences used for qPCR analysis.

Gene	Forward Sequence (5′-3′)	Reverse Sequence (5′-3′)
β-Actin	GGCTGTATTCCCCTCCATCG	CCAGTTGGTAACAATGCCATGT
TNFα	CTGAACTTCGGGGTGATCGG	GGCTTGTCACTCGAATTTTGAGA
IL-12p40	TGGTTTGCCATCGTTTTGCTG	ACAGGTGAGGTTCACTGTTTCT
TGFβ	AAAACAGGGGCAGTTACTACAAC	TGGCAGATATAGACCATCAGCA
IL-10	CTTACTGACTGGCATGAGGATCA	GCAGCTCTAGGAGCATGTGG
iNOS	GTTCTCAGCCCAACAATACAAGA	GTGGACGGGTCGATGTCAC
ARG-1	TTGGGTGGATGCTCACACTG	GTACACGATGTCTTTGGCAGA
IL-1β	GAAATGCCACCTTTTGACAGTG	TGGATGCTCTCATCAGGACAG
IL-6	TGAACAACGATGATGCACTTGCAG	TAGCCACTCCTTCTGTGACTCCAG

## Data Availability

The raw data used to support the findings of this study are available from the corresponding author upon request.

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
