# Peer review of "IL-21/IL-21R Promotes the Pro-Inflammatory Effects of Macrophages during C. muridarum Respiratory Infection"

_ijms, 2023, doi:10.3390/ijms241612557_

Round 1

Reviewer 1 Report

In the current manuscript authors have done a scientifically sound job of elucidating the mechanism of by which IL-21/IL-21R exacerbates chlamydia respiratory infection by promoting the proinflammatory effect of macrophages. The experiments were scientifically sound and authors have dived deep into mechanism. I do not find any major issues with the paper. 

However, I have one comment on the Figure 1F and 6C. Figure 1F is missing scale. Figure 6C has shown magnification but ideally should have a scale. 

Reviewer 2 Report

The manuscript entitled “IL-21/IL-21R promotes the pro-inflammatory effects of macrophages during C. muridarum respiratory infection", is well-written with an extensive discussion and addresses the main objectives of the study. It provides valuable insights into the mechanism by which IL-21/IL-21R exacerbates chlamydia respiratory infection through the promotion of proinflammatory Mφ responses. However, there are several key points that I believe are important to highlight and improvements can be made, that may benefit from further clarification or revision of the article (ijms-2528197).

The strength of the study includes the use of a combination of in vivo experiments with IL-21R-/- mice and in vitro experiments using C. muridarum-stimulated RAW264.7 cells to obtain comprehensive insights into the role of IL-21/IL-21R in regulating macrophage responses during chlamydial infection. Also, the adoptive transfer experiments using bone marrow-derived macrophages from IL-21R-/- mice and WT mice. This approach helps to demonstrate the specific contribution of IL-21/IL-21R in regulating the proinflammatory effects of Mφ during chlamydial infection along with integration of transcriptomic analysis. In contrast of strength of the manuscript, there are several limitations of the study, including, the study relies on a mouse model, which may not fully represent human chlamydial lung infection and the response of human macrophages to IL-21/IL-21R signaling. Additionally, the in vitro experiments may not fully capture the complexities of the in vivo lung environment, and therefore potentially limiting the generalizability of the findings, and last the mouse and human immune responses may differ, making it important to exercise caution when extrapolating results to human infections.

Furthermore, author need to correct the following comments which includes:

#line 41-44, The chlamydia strain used in this study (Chlamydia muridarum), which was isolated from laboratory mice and hamsters and primarily used as infection inoculum in mouse models for female genital and respiratory tract infections, why author has not used germ-free mice which will be important to elucidate this mechanism (host- Chlamydia infection), is there any specific reason not including germ-free mice.

The author has included and considers single pathogen focus (Chlamydia muridarum), this might limit generalizability to other pathogens, although it is partly discussed in line 491-492, “upregulation in the expression of 491 both IL-21 and IL-21R within lung tissue from infected C57BL/6 mice”. However, needs to discuss rational more in detail, also, other important cytokines (IL-21/IL-21R signaling) may have been overlooked, impacting the overall understanding, furthermore, why author has not cited or considered recent publication and how it varies and is it strain specific and cytokines specific mechanism (IL-27 signaling) https://doi.org/10.3390/microorganisms11030604 and

There are additional minor comments such as  

#line 59, 66,70,500, Staphylococcus aureus; Enterococcus faecalis, Klebsiella pneumoniae has not follow standard form, author need to maintain that standard form and uniformity in entire manuscript.

#line 97 pylmonary pathology, typo error “pulmonary”

#line100, Material. s and Methods, typo error

#line 50, As such, Professional phagocytes, the word “Professional” should me in lowercase.

#line 111, Mice were anesthetized with isoflurane anesthesia machine, “with isoflurane by using anesthesia machine, Author needs to correct it ”

#line223, 48h, #line 131,134, 2 hours, is not uniform,

# Representative H&E staining of Figure 6C, is not clear, needs to modify.
